# Performance and usability of cardiometabolic point of care devices in Nepal: A prospective, quantitative, accuracy study

**Marina Giachino** [1☉] *, **Beatrice Vetter** [2☉], **Sigiriya Aebischer Perone** [1], **Jorge César Correia** [3], **Berra Erkosar** [2], **Olivia Heller** [1], **Vijay Kumar Khanal** [4], **Bruno Lab** [1], **Zoltan Pataky** [3], **Sagar Poudel** [5], **Mamit Rai** [6], **Sanjib Kumar Sharma** [4]

1 Division of Tropical and Humanitarian Medicine, Geneva University Hospitals, Geneva, Switzerland, 2 FIND, Geneva, Switzerland, 3 Division of Endocrinology, Unit of Therapeutic Patient Education, WHO Collaborating Centre, Diabetology, Nutrition and Therapeutic Patient Education, Geneva University Hospitals, University of Geneva, Geneva, Switzerland, 4 B.P. Koirala Institute of Health Sciences (BPKIHS), Dharan, Nepal, 5 All India Institute of Medical Sciences, New Delhi, India, 6 Program for Early Detection and Management of Kidney Hypertension Diabetes and Cardiovascular Diseases (KHDC), Dharan, Nepal

☉ These authors contributed equally to this work.
* Marina.Giachino@hcuge.ch

**Data Availability Statement:** The full and final dataset of the clinical study is made available under the Supporting Information files.

## Abstract

Non-communicable diseases (NCDs), such as cardiovascular disease and diabetes, represent a serious global health concern. There is an urgent need for prompt diagnosis and effective monitoring at point of care, especially in low- and middle-income countries. Here we present the results of a study assessing the quantitative accuracy of two devices that may fit the target product profile for a cardiometabolic point-of-care device. This prospective, quantitative, accuracy study (NCT05257564) was conducted between March to May 2022, investigating the performance of the JanaCare Aina Blood Monitoring System (JCAina) and the Tascom SimplexTAS 101 device (TAS101) compared with local standard laboratory methods in rural Nepal. Using fingerstick capillary blood, cardiometabolic parameters were analysed using both devices. The quantitative accuracy was compared against a local laboratory reference assay. System usability was also assessed. For JCAina, the mean absolute biases (Bland-Altman analysis) for glucose, HbA1c and total cholesterol tests were -3.87 mg/dL (95% CI: -7.52–-0.22), 1.34% (95% CI: 1.21–1.47), and -9.52 mg/dL (95% CI: -11.9–-7.2), respectively, corresponding to mean percentage biases of 2.0%, 18.5%, and -6.4%. These indicate clinically small (<10% biases) differences from laboratory results for glucose and cholesterol, and a moderate (10–20%) positive bias for HbA1c. For TAS101, the mean absolute biases for glucose, HbA1c, total cholesterol and creatinine tests were 18.7 mg/dL (95% CI: 15.8–21.5), -0.2% (95% CI: -0.26–-0.14), 29.8 mg/dL (95% CI: 27.0–32.6), and -0.02 mg/dL (95% CI: -0.05–0.01), respectively, corresponding to mean percentage biases of 12.1%, -2.6%, 15.8%, and -4.5%. These indicate clinically small differences for HbA1c and creatinine, and moderate positive biases for glucose and cholesterol. Both systems exhibited usability challenges. The JCAina and TAS101 point-of-care cardiometabolic devices were shown to have promising accuracy in environmental conditions such as in Nepal, though improvements are still needed for some parameters and for ease of use.

**Funding:** This study was funded by FIND (www.
finddx.org), through a grant from the International
Committee of the Red Cross and the German
Federal Ministry of Education and Research
(NCT05257564; grant numbers: KFW-TBBU02;
ICR-NCDS01). The study or authors were not
funded by any commercial companies, no authors
received a salary or other funding from commercial
companies, and the funders had no role in the
study design, data collection and analysis, decision
to publish, or the preparation of the manuscript.

**Competing interests:** Marina Giachino, Zoltan
Pataky, Olivia Heller, Bruno Lab, Jorge César
Correia, Sanjib Kumar Sharma, Vijay Kumar
Khanal, Sagar Poudel, Mamit Rai declared that no
competing interests exist. I have read the journal's
policy and the following authors: Beatrice Vetter,
Berra Erkosar have the following competing
interests: they are both employees of FIND. As for
the author: Sigiriya Aebischer Perone, she has the
following competing interests: is an employee of
ICRC, the International Committee of the Red
Cross.

**Trial registration:** NCT05257564 (ClinicalTrials.gov).

## Introduction

Non-communicable diseases (NCDs), such as cardiovascular disease (CVD) and diabetes mellitus (DM), represent a serious global health concern, causing an estimated 41 million deaths every year, equivalent to 74% of all disease-related deaths worldwide [1]. Of these deaths, the majority (32 million; 77%) occur in low- and middle-income countries (LMICs), and half are associated with either CVD (17.9 million; 44%) or DM (2.0 million, including those with diabetic kidney disease; 5%) [1]. Screening, timely-diagnosis and treatment form the basis of an effective response to NCDs [1].

To facilitate diagnosis, management and follow-up of patients in remote and rural areas of LMICs, there is an urgent need to ensure that primary healthcare facilities are capable of prompt diagnosing and effective monitoring NCDs at the point of care.

To address this need, FIND, compiled a landscape of point-of-care devices capable of testing basic cardiometabolic clinical chemistry parameters, potentially suitable for supporting the diagnosis and monitoring of cardiometabolic diseases [2]. These devices measure parameters recommended by the World Health Organization for the detection of cardiometabolic risk, including lipids/lipoproteins, glucose, glycated haemoglobin (HbA1c) and serum creatinine (associated with CVD, DM, and kidney disease) [2–4]. Measuring a combination of parameters on a single point-of-care device has the potential to facilitate a more comprehensive view of a patient's condition by reducing the risk to forgo necessary tests due to the complexity of managing multiple point-of-care devices and supplies [5, 6].

In consultation with clinical and laboratory experts, FIND developed a target product profile (TPP) for cardiometabolic point-of-care devices intended for use in primary healthcare centres in LMICs, defining both minimal and optimal requirements (41 in total) [7]. Besides measuring the key cardiometabolic parameters, devices should be suitable for use in primary health care in low resource settings, capable of operating at environmental temperatures of ≥35˚C, be suitable for use by minimally trained healthcare workers, and involve the use of self-contained cartridges or strips without the need for additional reagents [7]. Using the TPP requirements to assess devices in the current landscape [2], two devices were identified that met several of these criteria (and particularly TPP characteristic number 1–5, 10, 12, 23, 24 and 26, describing requirements of the intended use, target setting and operators, device design, environmental stability, training time, test menu, test cartridges and need for additional consumable [7]): the JanaCare Aina Blood Monitoring System (JCAina) [8, 9] and the Tascom SimplexTAS 101 device (TAS101) [10, 11]. The JCAina device measures glucose, HbA1c, and total cholesterol (creatinine testing is in development) by way of dry test strips using fingerstick capillary whole blood [2, 8, 9]. While the glucose and total cholesterol tests are single-step processes, the HbA1c is a multi-step process involving buffer incubation before applying to the test strip [2, 8, 9]. The JCAina device can operate at up to 40˚C, something which is critical for many LMICs, and most reagents do not require refrigeration [2, 8, 9]. The TAS101 device measures glucose, HbA1c, total cholesterol and creatinine, by way of sample cartridges using fingerstick capillary whole blood [2, 10, 11]. The same procedure is followed for each test, and the device has an optional high temperature mode (up to 40˚C) [2, 10, 11].

Available manufacturer data indicates high testing accuracy for both systems [9, 11]. However, as with many point-of-care devices, their accuracy has not been rigorously evaluated in

their intended use setting and when operated by the intended users. Therefore, their accuracy in primary healthcare settings and routine use is unknown. These data are critical to enable the transition from laboratory testing to point-of-care testing and to ensure consistent patient management at primary health care facilities. Here we present the results of a study conducted to assess the quantitative accuracy of the JCAina and TAS101 devices in their intended use setting, compared with local standard-of-care laboratory testing.

## Materials and methods

### Study design

This was a prospective, quantitative, accuracy study investigating the performance of two clinical chemistry cardiometabolic point-of-care devices compared with local standard of care laboratory methods in Nepal. The study was conducted between 9 March 2022 and 26 May 2022 at the Dhulabari primary healthcare centre and Kakarvitta health post in rural Jhapa district (Eastern Nepal), as part of the project "Increasing awareness, detection and management of diabetes, hypertension, chronic kidney and cardiovascular diseases in Eastern Nepal" (KHDC) [12]. The study was embedded in the KHDC-Nepal program, which has the objective to improve the health of the Nepalese population in Eastern Nepal through early detection and management of chronic noncommunicable diseases. The program runs community awareness sessions and door-to-door visits to encourage members of the community to attend a screening visit at selected primary health facilities, health posts or at a pre-designated location. During this visit, people are screened for hypertension, diabetes, chronic kidney and cardiovascular disease, including (but not limited to) testing for glucose, lipids, HbA1c and creatinine, therefore the participants of this study are the intended use population for the point of care devices.

### Participants

Participants who attended either of the study sites as part of the KHDC programme (for screening or follow-up appointments) [12] were eligible for inclusion in this study if they were 20 years or older, able and willing to provide informed consent, and had haemoglobin levels ≥8 g/dL. Participants were excluded from the study if they were unable to provide sufficient capillary or venous whole blood sample for all tests, had haemoglobin levels <8 g/dL, or were attending the Dhulabari primary healthcare centre or Kakarvitta health post for reasons other than the KHDC programme.

### Interventions and procedures

Upon enrolment, fingerstick capillary blood was collected from all eligible participants and analysed immediately using the JCAina and TAS101 devices. Study staff were trained on the use of the JCAina and TAS101 by the manufacturers via video call with on-site supervision by the study manager and the principal investigator. Both devices were installed closely in line with the manufacturers' instructions and reagents kept at the indicated storage temperatures (2–8°C for TAS101 and 2–30°C for JCAina total cholesterol, 1–30°C for JCAina glucose, 2–8°C for JCAina HbA1c). Quality control testing for each parameter was performed by the study's laboratory staff, once per week using the manufacturers' control solutions and guidelines with the target values to be reached. For reference laboratory testing, samples for plasma, whole blood and serum testing were collected immediately after fingerstick whole blood collection (for investigational device testing). Plasma in K3-EDTA collection tubes was prepared from venous blood by centrifugation within 30 minutes of sample collection to minimize

glycolysis [13] and serum was prepared by centrifugation after the blood samples had been left to clot for up to 60 minutes. All samples were then stored at 2–8˚C until transferred the same day to the B.P. Koirala Institute of Health Sciences laboratory and tested the same day or the next morning, using one Roche Cobas c311 analyser [14], according to the manufacturer's instructions and performing all local quality control/calibration standard operating procedures. Before processing the samples at the reference laboratory, they were stored in cooling boxes with temperature monitoring throughout the transport and then stored at 2–8˚C.

## Outcomes

The primary outcome was the accuracy of quantitative measurements of glucose, HbA1c, total cholesterol and creatinine as measured by the JCAina and TAS101 multiparameter point-of-care devices compared with a local laboratory reference assay. Estimates of agreement (Passing Bablok regression), fixed bias (mean difference), and limits of agreement (Bland-Altman analysis) were assessed for each test. Secondary outcomes included measures of operational characteristics (the proportion of invalid tests per parameter for each device) and system usability measured using the System Usability Scale [15]. This scale employs a set of 10 questions using a Likert scale from 1–5 (1: 'strongly disagree'; 5: 'strongly agree') and has a maximum score of 100 [15]. One point is deducted from every odd-numbered question, and for every even-numbered question the score is subtracted from five, then the total is multiplied by 2.5 to obtain the final value out of 100 [15]. System Usability Scale scores are then rated as follows: <51 is considered 'awful'; 51–68 'poor'; 68 'okay'; >68–80.3 'good'; and >80.3 'Excellent' [15].

## Sample size and statistical analysis

Available data from the manufacturers on the performance of the JCAina and TAS101 cite Pearson correlation coefficients greater than 0.98 for all primary outcome markers compared with a reference laboratory method. Precision estimates (coefficient of variation [%CV]) ranged from 1.9–3.9% and 1.8–4.1% for the JCAina and TAS101 devices, respectively, dependent on the markers and their concentration. Using these assumptions, it was estimated that a sample size of 337 participants would be needed to detect a statistically significant difference of ±1.5% (with 80% power using a two-tailed t-test and a 5% significance level) in primary outcome markers compared with the laboratory reference assay. To improve precision and account for the possibility of lower field performances, a sample size of 400 participants was selected.

All statistical and graphical analyses were performed using R (version 4.2.0) and ggplot2 Library, as well as the SAS statistical analysis software suite [16–18]. For the analysis of the primary outcome, point estimates (and 95% confidence intervals [CI]) of agreement between the JCAina/TAS101 tests and laboratory tests were calculated using the Passing Bablok regression method, where a slope of 1 (representing proportional difference) and an intercept of 0 (representing systematic difference) constitutes perfect agreement [19]. Systemic deviations of JCAina/TAS101 tests from laboratory tests (mean bias) were calculated using Bland-Altman plots in both absolute and relative units. All other data are presented using descriptive statistics.

## Ethics statement

This study was conducted in accordance with the protocol (available under ClinicalTrials.gov), the ethical principles of the Declaration of Helsinki, and all other applicable laws and regulations. The Institutional Review Board in Nepal (Nepal Health Research Council) reviewed and approved the study documents (including the protocol) before the study was initiated and the

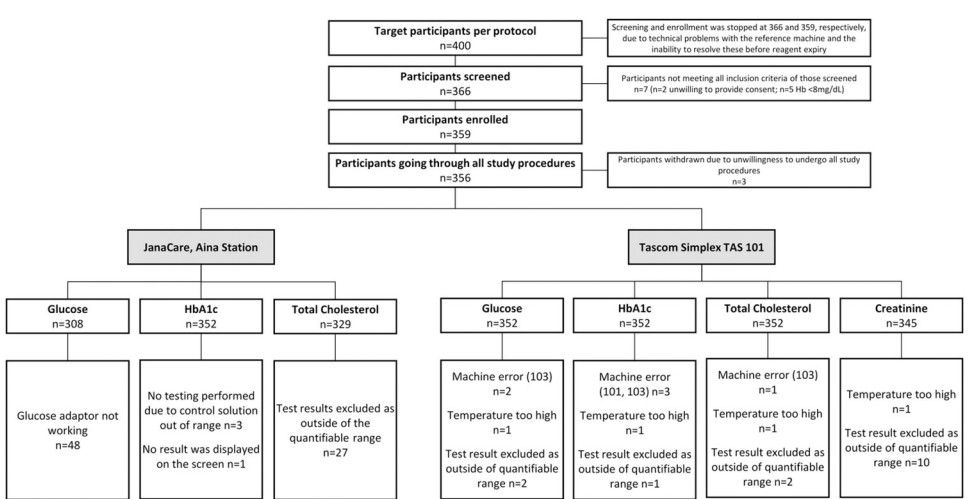

**Fig 1. Patient and testing disposition.** Hb, haemoglobin; HbA1c, glycated haemoglobin.

Geneva Cantonal Commission of Ethics and Research were provided with the same documents and emitted a positive Advisory Opinion. All participants provided written informed consent before participation in the study (prior to haemoglobin screening).

## Results

In total, 366 participants were screened, 359 met the eligibility criteria, and 356 underwent all study procedures (Fig 1). Recruitment was terminated prior to reaching the target of 400 participants due to technical problems with the laboratory equipment, which could not be addressed before the expiry date of some of the point-of-care devices' reagents.

### Baseline demographics and characteristics

Participant baseline demographics and characteristics are presented in **Table 1**. Overall, participant characteristics were similar between study sites, with a mean age of 51.8 (±14.1) years, a mean body mass index of 25.5 (±4.4) kg/m², and the majority of participants being female (57.9%), non-smokers (69.7%), non-drinkers of alcohol (76.7%), and just over half of participants had a diagnosis of at least one NCD (50.56%).

### Outcomes

**Test accuracy.** For the JCAina device, Passing Bablok analysis showed slopes of 0.87 (95% CI: 0.84–0.90), 1.10 (95% CI: 1.02–1.19), and 1.08 (95% CI: 1.02–1.15), for glucose, HbA1c and total cholesterol testing, respectively. Intercepts for glucose, HbA1c and total cholesterol were 20.4 mg/dL (95% CI: 16.1–24.4; upper measuring range limit: 600 mg/dL), 0.61% (95% CI: 0.05–1.09; upper measuring range limit: 15.0%), and -23.34 mg/dL (95% CI: -33.7–-13.9; upper measuring range limit: 400 mg/dL), respectively (where a value of 0 would indicate perfect agreement between tests and laboratory values) (Fig 2). The mean absolute biases (Bland-Altman analysis) for glucose, HbA1c and cholesterol tests were -3.87 mg/dL (95% CI: -7.52–0.22), 1.34% (95% CI: 1.21–1.47), and -9.52 mg/dL (95% CI: -11.9–-7.2), respectively, corresponding to mean percentage biases of 2.0% (95% CI: 0.4–3.5), 18.5% (95% CI: 17.0–20.1), and -6.4% ((95% CI:-7.9 –-5.0); Fig 3). Overall, these data indicate a clinically slight (<10% bias) difference between JCAina and laboratory results for glucose and cholesterol, within or similar

**Table 1. Patient baseline demographics and characteristics.**

| | Dhulabari public healthcare facility (n = 175) | Kakarvitta health post (n = 181) | Total (N = 356) |
|---|---|---|---|
| Sex, n (%) | | | |
| Female | 100 (57.1) | 106 (58.6) | 206 (57.9) |
| Male | 75 (42.9) | 75 (41.4) | 150 (42.1) |
| Age, years, mean (±SD) | 52.9 (±14.4) | 50.8 (±13.8) | 51.8 (±14.1) |
| Fasting status, n (%) | | | |
| Yes* | 8 (4.6) | 23 (12.7) | 31 (8.7) |
| No | 167 (95.4) | 158 (87.3) | 325 (91.3) |
| Smoking status, n (%) | | | |
| Smoker† | 34 (19.4) | 50 (27.6) | 84 (23.6) |
| Former smoker | 11 (6.3) | 11 (6.1) | 22 (6.2) |
| Non-smoker | 130 (74.3) | 118 (65.2) | 248 (69.7) |
| NR | 0 | 2 (1.1) | 2 (0.6) |
| Alcohol intake, n (%) | | | |
| Once a day | 6 (3.4) | 11 (6.1) | 17 (4.8) |
| Once a week | 13 (7.4) | 19 (10.5) | 32 (9.0) |
| Once a month | 12 6.9) | 20 (11.0) | 32 (9.0) |
| None | 144 (82.3) | 129 (71.3) | 273 (76.7) |
| NR | 0 | 2 (1.1) | 2 (0.6) |
| BMI, kg/m², mean (±SD) | 25.2 (±4.3) | 25.8 (±4.6) | 25.5 (±4.4) |
| KHDC diagnosis, n (%) | | | |
| Hypertension | 75 (42.9) | 61 (33.7) | 136 (38.2) |
| Diabetes mellitus | 66 (37.7) | 43 (23.8) | 109 (30.6) |
| Cardiovascular disease | 2 (1.1) | 3 (1.7) | 5 (1.4) |
| Chronic kidney disease | 1 (0.6) | 4 (2.2) | 5 (1.4) |
| None | 65 (37.1) | 86 (47.5) | 151 (42.4) |
| NR | 10 (5.7) | 14 (7.7) | 24 (6.7) |
| Concomitant medication, n (%) | | | |
| BP lowering medication | 66 (37.7) | 61 (33.7) | 127 (35.7) |
| Oral antiglycaemic agents | 50 (28.6) | 38 (21.0) | 88 (24.7) |
| Traditional local medication‡ | 2 (1.1) | 0 | 2 (0.6) |
| Lipid lowering medication | 8 (4.6) | 17 (9.4) | 25 (7.0) |
| Other glucose lowering medication | 3 (1.7) | 7 (3.9) | 10 (2.8) |
| Other | 1 (0.6) | 13 (7.2) | 14 (3.9) |
| None | 84 (48.0) | 95 (52.5) | 179 (50.3) |

*No food or drink other than water for 8 hours; †Within the past year; ‡such as Hoodia, Gymnema, or Aloe.

BMI, body mass index; BP, blood pressure; NR, not recorded; SD, standard deviation.

to the recommended thresholds for clinically important differences and acceptable analytical performance according to the 2014 Royal College of Pathologists of Australasia (RCPA) and 1992 Clinical Laboratory Improvement Amendments (CLIA) requirements for analytical quality (± 10% for glucose; ± 6–10% for cholesterol) [20, 21]. For HbA1c, a moderate (10–20%) positive bias was observed, which exceeded the RCPA/CLIA recommended thresholds (± 6–8%) [20, 21].

For the TAS101 device, Passing Bablok analysis showed slopes of 1.05 (95% CI: 1.02–1.08), 0.84 (95% CI: 0.80–0.88), 1.30 (95% CI: 1.24–1.38), and 1.15 (95% CI: 1.03–1.32), for glucose, HbA1c, cholesterol and creatinine testing, respectively. Intercepts for glucose, HbA1c,

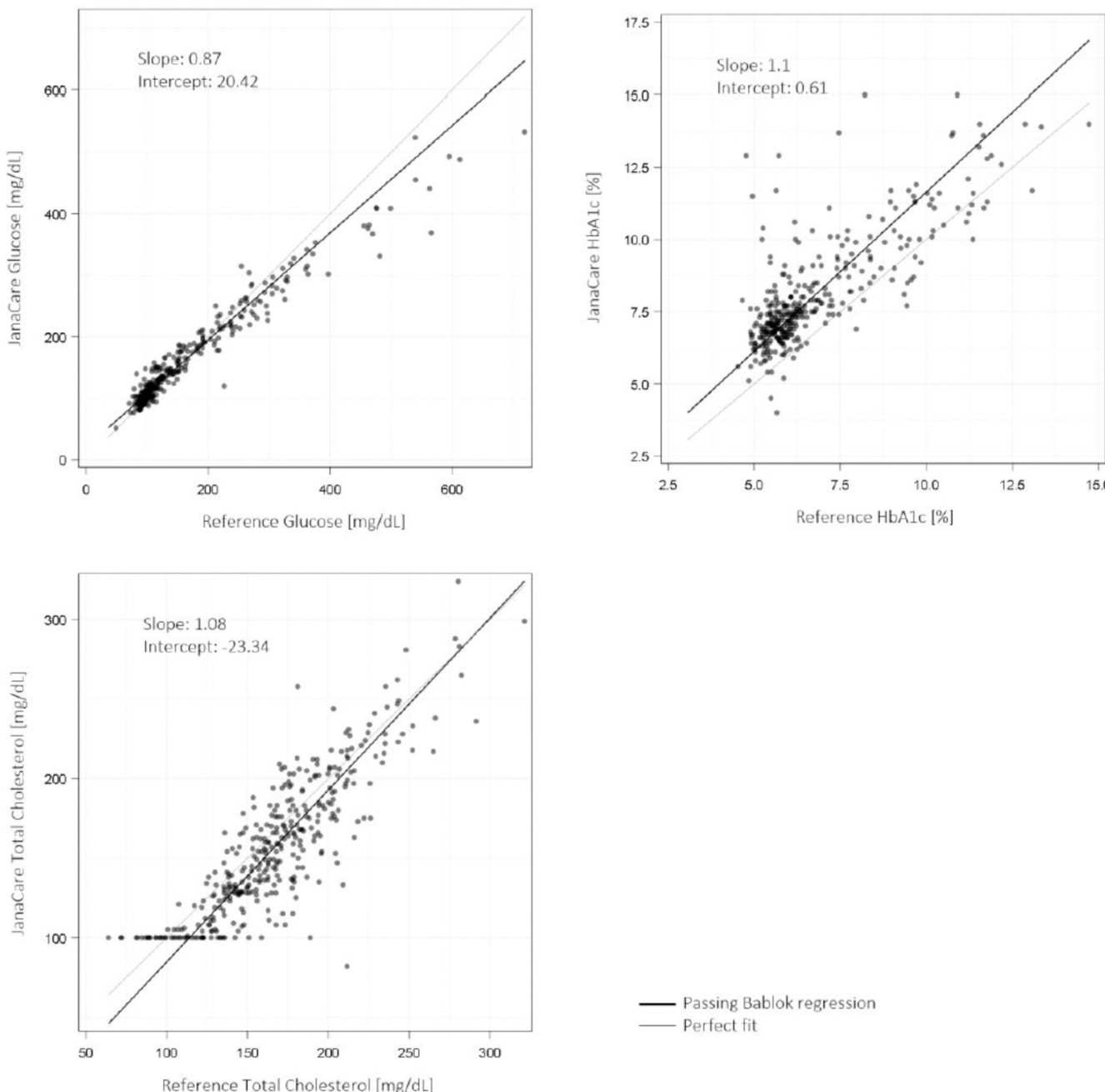

**Fig 2. Passing Bablok analysis of JCAina tests (glucose, HbA1c, total cholesterol).** HbA1c, glycosylated haemoglobin; JCAina, JanaCare Aina Blood Monitoring System.

cholesterol and creatinine were 7.78 mg/dL (95% CI: 3.62–12.02; upper measuring range limit: 700 mg/dL), 0.85% (95% CI: 0.60–1.08; upper measuring range limit: 15.0%), -22.1 mg/dL (95% CI: -33.7–-11.1; upper measuring range limit: 500 mg/dL), and -0.17 mg/dL (95% CI: -0.29–-0.07; upper measuring range limit: 25.0 mg/dL), respectively (Fig 4). The mean absolute biases (Bland-Altman analysis) for glucose, HbA1c, total cholesterol and creatinine tests were 18.7 mg/dL (95% CI: 15.8–21.5), -0.2% (95% CI: -0.26–-0.14), 29.8 mg/dL (95% CI: 27.0–32.6), and -0.02 mg/dL (95% CI: -0.05–0.01), respectively, corresponding to mean percentage biases

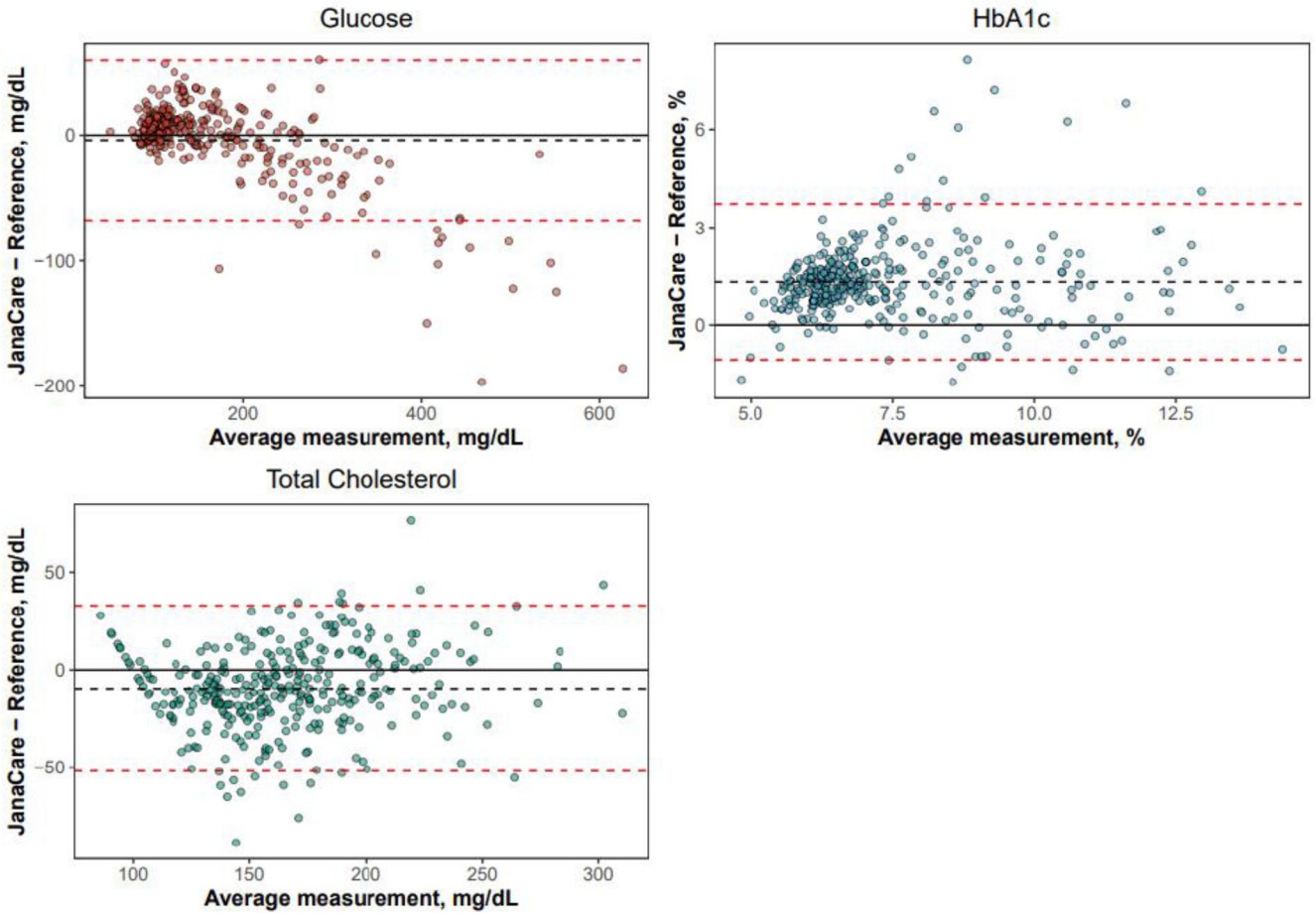

**Fig 3. Bland-Altman analysis of JCAina tests (glucose, HbA1c, total cholesterol).** HbA1c, glycosylated haemoglobin; JCAina, JanaCare Aina Blood Monitoring System.

of 12.1% (95% CI: 10.8–13.5), -2.6% (95% CI: -3.4–-1.9), 15.8% (95% CI: 14.5–17.1), and -4.5% ((95% CI: -8.4–-0.7); Fig 5). Overall, these data indicate clinically small differences between TAS101 and laboratory results for HbA1c and creatinine (<10%; both within RCPA/CLIA recommended thresholds of ± 6–8% and ± 8–10%, respectively), and a moderate positive bias (10–20%, exceeding RCPA/CLIA recommended thresholds) for glucose and total cholesterol results [20, 21].

**Operational characteristics.** For the JCAina device, three technical problems were encountered: one blank screen after insertion of the glucose test, and two HbA1c tests where no results or error codes were displayed. In addition, an adaptor failure resulted in 48 glucose tests not being performed, and three HbA1c tests could not be performed as the control solution test was out of range.

For the TAS101 device, four initial test failures (no results obtained; no error code provided) occurred with glucose, nine with HbA1c, four with cholesterol and two with creatinine tests; repeat tests also failed in three, four, two and one cases, respectively. All error codes reported were related to the sample collection device (e.g., insufficient/excess sample volume, wrong sample/device used, missing sample).

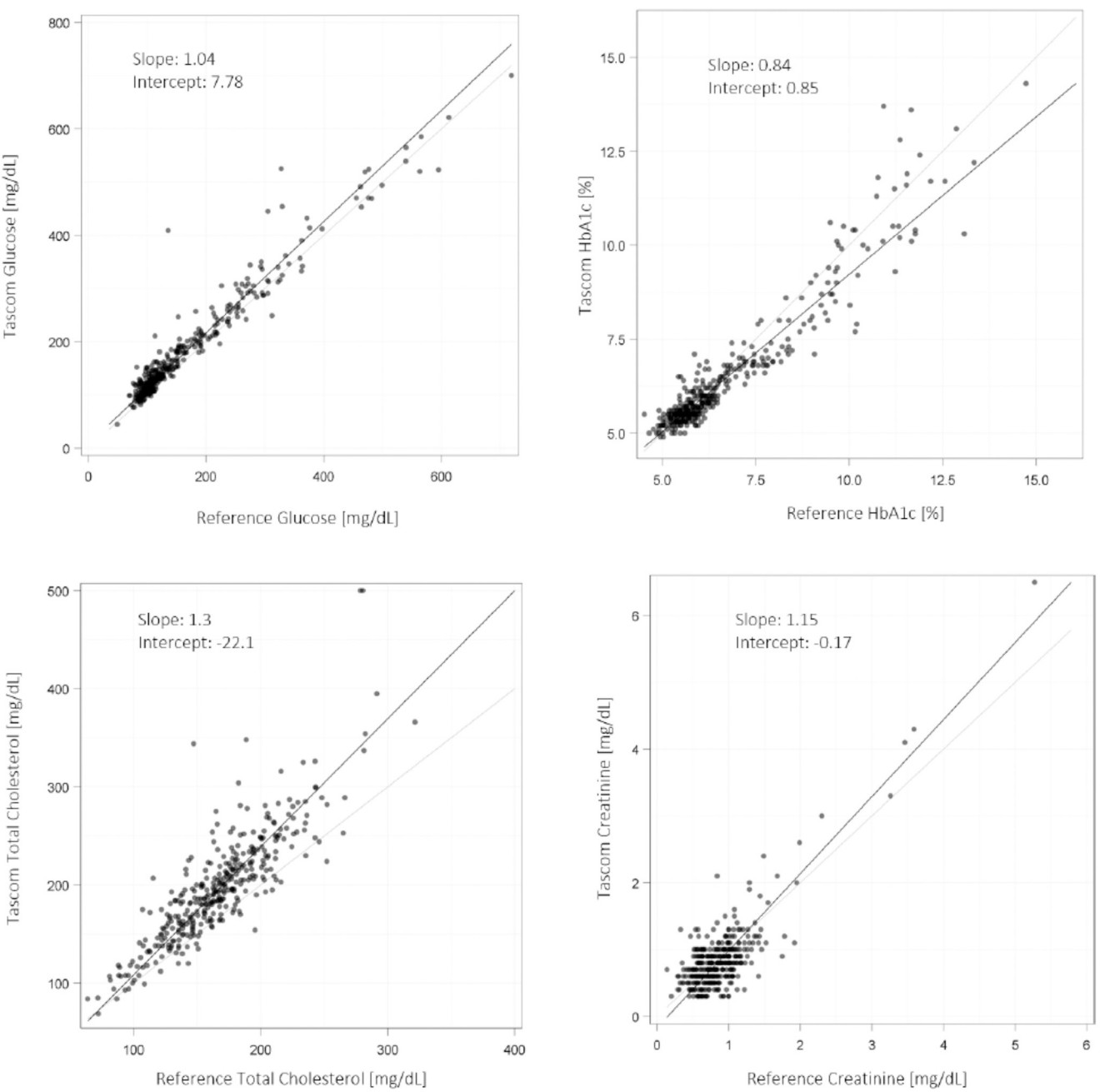

**Fig 4. Passing Bablok analysis of TAS101 tests (glucose, HbA1c, total cholesterol, creatinine).** HbA1c, glycosylated haemoglobin; TAS101, Tascom SimplexTAS 101 device.

**System usability.** The System Usability Scale was completed by five staff members who were trained in the use of the devices and were actively using them. Full details of their responses are presented in **Table 2**.

For the JCAina device, the mean System Usability Score was 58 ('poor') as a result of variable, but critical ratings, primarily related to ease of use. While users felt confidence in using the system, they did not feel it was cumbersome to use, and did not feel that they needed technical support in using it (average ratings of 4.2/5, 2.4/5 and 2.2/5, respectively). There was

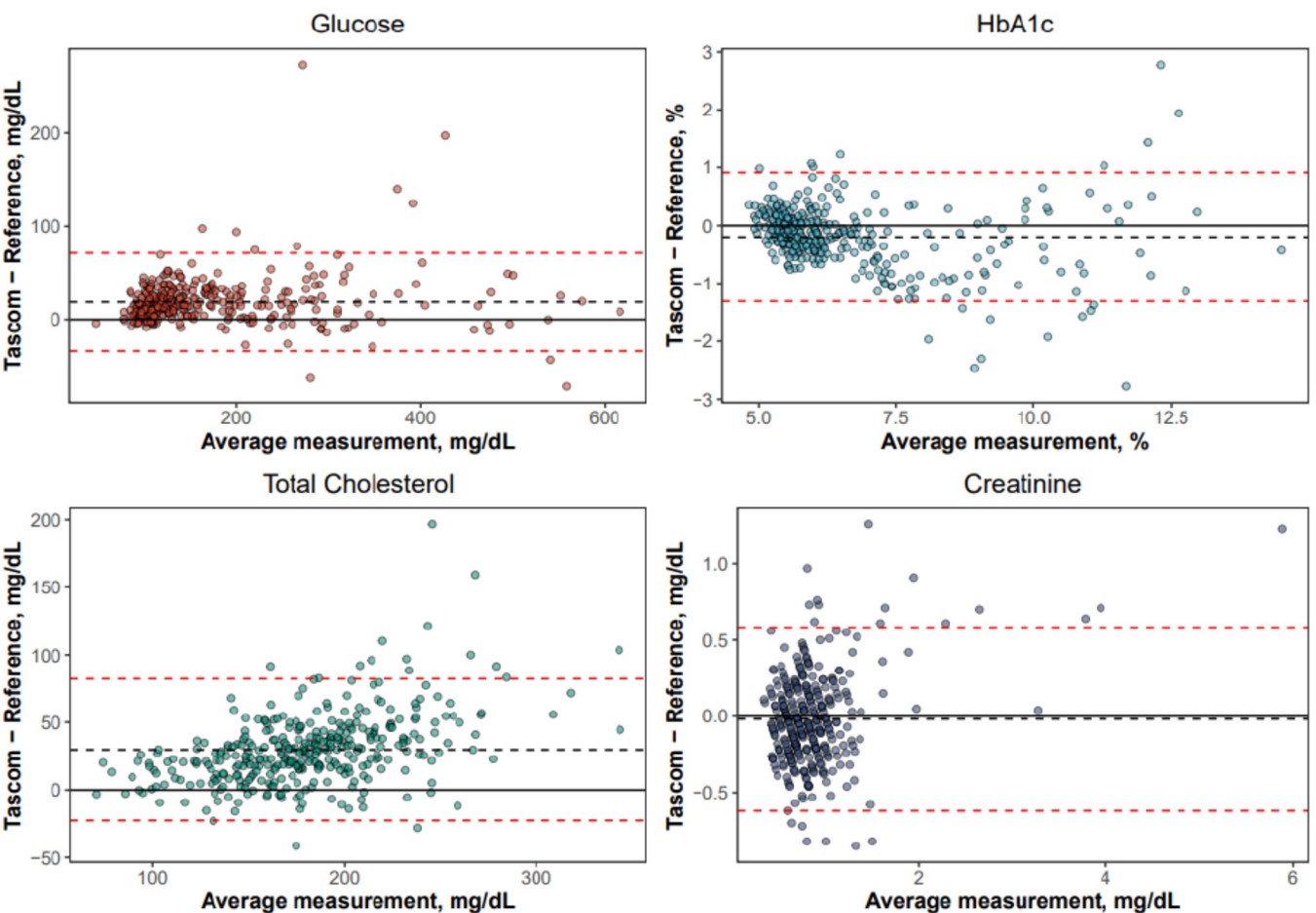

**Fig 5. Bland-Altman analysis of TAS101 tests (glucose, HbA1c, total cholesterol, creatinine).** HbA1c, glycosylated haemoglobin; TAS101, Tascom SimplexTAS 101 device.

however a strong agreement that the system required a lot of training before use (average rating 4.2/5), and three of the users felt that there was too much inconsistency in the system (individual ratings of 3/5).

For the TAS101 device, the mean System Usability Score was 71.5 ('good'), with three out of the five users reporting 'good' ratings overall. The ratings profile was similar to the JCAina device, but with better ratings overall. Users felt confident in using the system and thought it was easy to use (average ratings of 4.6/5 and 4.4/5, respectively). There was again a strong agreement that the system required a lot of training before use (average rating 4.4/5), and the two users who rated the system as 'poor' felt that there was too much inconsistency in the system (individual ratings of 3/5).

## Discussion

In this prospective, quantitative, accuracy study, two multiparameter cardiometabolic point-of-care diagnostic devices, which closely matched the TPP requirements (compared with other devices in the landscape) [2, 7], were compared with local standard-of-care laboratory tests in a real-world, primary healthcare care setting in Nepal. The main aim of the study was to understand the quantitative differences that clinicians may expect when implementing point-

**Table 2. Summary of system usability scale results for JCAina and TAS101.**

| | Strongly disagree = 1, Strongly agree = 5 | | | | | | |
|---|---|---|---|---|---|---|---|
| **JCAina System Usability Scale** | **User 1** | **User 2** | **User 3** | **User 4** | **User 5** | **Average** | **Calc.** |
| 1. I think that I would like to use this system frequently | 4 | 4 | 4 | 1 | 4 | 3.4 | x-1 |
| 2. I found the system unnecessarily complex | 2 | 3 | 4 | 2 | 4 | 3.0 | 5-x |
| 3. I thought the system was easy to use | 4 | 4 | 4 | 3 | 3 | 3.6 | x-1 |
| 4. I think that I would need the support of a technical person to be able to use the system | 3 | 1 | 3 | 1 | 3 | 2.2 | 5-x |
| 5. I found the various functions in this system were well integrated | 4 | 3 | 3 | 1 | 4 | 3.0 | x-1 |
| 6. I thought there was too much inconsistency in this system | 2 | 3 | 3 | 1 | 3 | 2.4 | 5-x |
| 7. I would imagine that most people would learn to use this system very quickly | 4 | 3 | 4 | 1 | 4 | 3.2 | x-1 |
| 8. I found the system very cumbersome to use | 2 | 2 | 2 | 3 | 3 | 2.4 | 5-x |
| 9. I felt very confident using the system | 5 | 4 | 4 | 4 | 4 | 4.2 | x-1 |
| 10. I needed to learn a lot of things before I could get going with this system | 4 | 4 | 4 | 5 | 4 | 4.2 | 5-x |
| **Total System Usability Scale score** | **70** | **62.5** | **57.5** | **45.0** | **55.0** | **58.0** | x2.5 |
| System Usability Scale grade | **B** | **D** | **D** | **F** | **D** | **D** | |
| System Usability Scale adjective rating | Good | Poor | Poor | Awful | Poor | Poor | |
| **TAS101 System Usability Scale** | **User 1** | **User 2** | **User 3** | **User 4** | **User 5** | **Average** | **Calc.** |
| 1. I think that I would like to use this system frequently | 4 | 4 | 4 | 5 | 4 | 4.2 | x-1 |
| 2. I found the system unnecessarily complex | 1 | 2 | 2 | 1 | 2 | 1.6 | 5-x |
| 3. I thought the system was easy to use | 5 | 3 | 5 | 5 | 4 | 4.4 | x-1 |
| 4. I think that I would need the support of a technical person to be able to use the system | 2 | 2 | 2 | 1 | 3 | 2.0 | 5-x |
| 5. I found the various functions in this system were well integrated | 4 | 4 | 4 | 4 | 4 | 4.0 | x-1 |
| 6. I thought there was too much inconsistency in this system | 1 | 3 | 2 | 1 | 3 | 2.0 | 5-x |
| 7. I would imagine that most people would learn to use this system very quickly | 5 | 3 | 4 | 3 | 3 | 3.6 | x-1 |
| 8. I found the system very cumbersome to use | 2 | 2 | 2 | 3 | 2 | 2.2 | 5-x |
| 9. I felt very confident using the system | 5 | 4 | 5 | 5 | 4 | 4.6 | x-1 |
| 10. I needed to learn a lot of things before I could get going with this system | 5 | 4 | 4 | 5 | 4 | 4.4 | 5-x |
| **Total System Usability Scale score** | **80** | **62.5** | **75** | **77.5** | **62.5** | **71.5** | x2.5 |
| System Usability Scale grade | **B** | **D** | **B** | **B** | **D** | **B** | |
| System Usability Scale adjective rating | Good | Poor | Good | Good | Poor | Good | |

Calc., calculations; JCAina, JanaCare Aina Blood Monitoring System; TAS101, Tascom SimplexTAS 101 device.

of-care testing in primary healthcare facilities, compared with central laboratory testing available in their setting. Overall, both the JCAina and TAS101 devices showed promising accuracy, with the majority of parameters measured with low bias (<10% difference versus laboratory results), within or similar to the recommended thresholds for clinically important differences and acceptable analytical performance according to RCPA/CLIA guidelines [20, 21]. This indicates they could be suitable for point-of-care testing in a primary healthcare setting. However, some parameters did show positive bias (>10% difference versus laboratory results) that exceeded RCPA/CLIA thresholds, such as HbA1c with JCAina and glucose and total cholesterol with TAS101. Measuring HbA1c, for example, has clinical implication in the management and prevention of diabetes complication. False low measurements can erroneously imply that diabetes is under control, while false high measurements can justify stricter control measures that would increase the risk of hypoglycemia and its aftereffects. As such, it is critical that the level of acceptable bias is established before making decisions about point-of-care device implementation and use. For example, while we indicate that JCAina shows 'moderate' bias for HbA1c testing, in relative terms this is approximately two-to-three-fold higher than the estimated bias for other dedicated diabetes point-of-care analysers (Afinion [Abbott] and

DCA Vantage [Siemens]) reported in a community-based study, while the TAS101 shows approximately two-fold less bias [22]. Given that the Afinion and DCA Vantage devices have been shown to meet the generally accepted performance criteria for HbA1c in a separate laboratory-based study [23], the information presented here could be useful in making decisions on the use of JCAina or TAS101 in specific practices based on the most common testing procedures required.

The International Federation of Clinical Chemistry recommends that "Device performance claims from the package insert should correlate with those obtained from verification studies that are performed by the [point-of-care testing] POCT personnel" [24]. This is critically important, considering variable field performance of point-of-care tests, [25–27] the lack of manufacturer-independent evaluations and the high number of on-market cardiometabolic POC devices [28].

The reasons for discrepancies versus laboratory results found in our study are unclear. For JCAina HbA1c testing it could be due to pre-analytical handling variability, as the test requires several manual steps and sample/buffer mixing [2, 8, 9]. For the TAS101 glucose and total cholesterol discrepancies, the reasons are less apparent, though it is likely due to a combination of technical variability with the instrument, user variability, and environmental factors. In particular, the manufacturer's instructions recommend installation of the TAS101 device away from direct sunlight or draughts with sufficient space around the machine [10], which was not always possible at one of the sites due to large windows providing sunlight access through much of the laboratory. This could have led to variations in the results dependent on the time of day and set up. We can also not exclude that the observed bias in glucose measurement is not a result of rapidly changing glucose levels in participants during the time of sample collection as the majority of participants had their samples taking in non-fasting status. However, the time between fingerstick and venous whole blood sample collection was kept to a minimum as participants immediately proceeded to venous blood collection after point-of-care fingerstick testing, so we would expect this source of potential variability to be minimal. Degradation of glucose through glycolysis in the reference sample could also be a potential source of positive bias observed in the study, however any effect was likely minimal (around 0.1 mmol/l or 1.8 mg/dL), based on findings of a previous study on glucose degradation kinetics in venous blood samples kept at room temperature for 30 min [29].

The coefficient of variation (CV) for the reference tests are another source of variability in this study. However, based on the instructions for use by the manufacturer, the Roche cobas tests for glucose, cholesterol and HbA1c used in this study had a CV (repeatability) of <2% and ≤4% for creatinine, suggesting that the effect was minimal with a CV <5% often considered excellent [30, 31].

Lastly, it is possible that difference in detection methods used by the investigational devices and the laboratory reference devices may have contributed to the observed difference. For glucose and total cholesterol, the investigation tests, as well as the reference tests were based on an enzymatic method. For HbA1c detection, JCAina used the boronate affinity method, TAS101 an enzymatic method and the reference test HbA1c detection was an immunoassay. For creatinine, the TAS101 used an enzymatic method and the reference test was based on the Jaffe method. While this may have contributed to the observed differences between the investigational tests and the reference tests, e.g., due to different risks of interfering substances [32], it highlights the challenges that implementers will face when transitioning from central laboratory to point-of-care testing as testing methods are likely to vary in real-life settings.

Both devices also had challenges in terms of usability, something identified in the TPP as a key aspect for successful implementation in primary healthcare [7]. While users mostly felt that most people would learn to use both systems quickly, they judged the learning required to

be substantial prior to first system use. These were not just limited to ease of use, device complexity or technical errors and or issues, but also acceptance by the users. This is something that manufacturers would be advised to take into consideration when optimising devices to ensure their long-term use. In addition, while all reagents were temperature monitored in this study, it should be noted that the JCAina device required some reagents to be stored at 2–8˚C [2, 8, 9], as well as for all reagents of TAS101 [2, 10, 11], which adds another layer of complexity (i.e., temperature/refrigeration monitoring) and an additional source of error (i.e., reagent quality). It is important to consider the potential influence of factors such as available infrastructure, complexity versus ease of use, and storage, as these would not be apparent in a controlled clinical trial or laboratory environment. This highlights the value of (i) conducting studies in a real-world setting, and (ii) having a comprehensive TPP available during development to identify key areas (beyond testing accuracy) that are important for making devices fit for purpose in LMICs.

In conclusion, the quantitative accuracy of the JCAina and TAS101 point-of-care cardiometabolic devices in intended use settings was shown to be promising, though investments are still needed to improve key aspects such as consistent performance across the whole portfolio of parameters, reagent storage temperature, as well as ease of use.

## Supporting information

**S1 Checklist. STARD checklist for accuracy studies.**
(PDF)

**S1 Dataset. Final study dataset.**
(XLSX)

## Acknowledgments

The authors would like to acknowledge the contributions of all participating staff at the Dhulabari primary healthcare centre and Kakarvitta health post in rural Jhapa district (Eastern Nepal), as well as all study participants. Editorial assistance in the preparation of the manuscript was provided by Stuart Wakelin, PhD.

## Author Contributions

**Conceptualization:** Marina Giachino, Beatrice Vetter, Sigiriya Aebischer Perone, Jorge César Correia, Olivia Heller, Bruno Lab, Zoltan Pataky, Sanjib Kumar Sharma.

**Data curation:** Marina Giachino.

**Formal analysis:** Berra Erkosar.

**Investigation:** Vijay Kumar Khanal, Sagar Poudel, Mamit Rai, Sanjib Kumar Sharma.

**Supervision:** Beatrice Vetter.

**Writing – original draft:** Marina Giachino, Beatrice Vetter.

**Writing – review & editing:** Marina Giachino, Beatrice Vetter, Sigiriya Aebischer Perone, Jorge César Correia, Berra Erkosar, Olivia Heller, Vijay Kumar Khanal, Bruno Lab, Zoltan Pataky, Sagar Poudel, Mamit Rai, Sanjib Kumar Sharma.

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
