## [Decision Letter · Decision Letter 0]

6 Mar 2024

PGPH-D-24-00096

Performance and usability of cardiometabolic point of care devices in Nepal: a prospective, quantitative, accuracy study

Dear Dr. Giachino,

Thank you for submitting your manuscript to PLOS Global Public Health. After careful consideration, we feel that it has merit but does not fully meet PLOS Global Public Health’s publication criteria as it currently stands. Therefore, we invite you to submit a revised version of the manuscript that addresses the points raised during the review process.

Please note that we have only been able to secure a single reviewer to assess your manuscript. We are issuing a decision on your manuscript at this point to prevent further delays in the evaluation of your manuscript. Please be aware that the editor who handles your revised manuscript might find it necessary to invite additional reviewers to assess this work once the revised manuscript is submitted. However, we will aim to proceed on the basis of this single review if possible.

The reviewer has queried a range of aspects of the study design; please ensure you address each of the reviewer's comments when revising your manuscript.

We look forward to receiving your revised manuscript.

Kind regards,

Hugh Cowley

Staff Editor

Journal Requirements:

1. Please send a completed 'Competing Interests' statement, including any COIs declared by your co-authors. If you have no competing interests to declare, please state "The authors have declared that no competing interests exist". Otherwise please declare all competing interests beginning with the statement "I have read the journal's policy and the authors of this manuscript have the following competing interests:"

2. We have noticed that you have uploaded Supporting Information files, but you have not included a list of legends. Please add a full list of legends for your Supporting Information files after the references list.

3. In the online submission form, you indicated that "The data will be made available by the corresponding author upon reasonable request". All PLOS journals now require all data underlying the findings described in their manuscript to be freely available to other researchers, either 1. In a public repository, 2. Within the manuscript itself, or 3. Uploaded as supplementary information.

Additional Editor Comments (if provided):

Reviewers' comments:

Reviewer's Responses to Questions

**Comments to the Author**

1. Does this manuscript meet PLOS Global Public Health’s publication criteria? Is the manuscript technically sound, and do the data support the conclusions? The manuscript must describe methodologically and ethically rigorous research with conclusions that are appropriately drawn based on the data presented.

Reviewer #1: Yes

2. Has the statistical analysis been performed appropriately and rigorously?

Reviewer #1: Yes

3. Have the authors made all data underlying the findings in their manuscript fully available (please refer to the Data Availability Statement at the start of the manuscript PDF file)?

Reviewer #1: No

4. Is the manuscript presented in an intelligible fashion and written in standard English?

Reviewer #1: Yes

5. Review Comments to the Author

Reviewer #1: Interesting manuscript addressing an important global need.

To improve scientific clarity, please address below comments

1. all figures are of a poor quality

2. has the protocol been published?

3. has the authors considered the relevance of learning curve?

4. what are the essential characteristics for the point of care devices as presented in TPP. Author mentioned 41 characterstics in total, are there any key characteristics? Would be good to know how these two devices have been chosen

5. how was the quality control carried out? By whom, using what standard to ensure the device is operating safety and accurately?

6. was there any study in terms of how methods and timing treating and analysing samples result in the discrepancies (biases) as observed in this study?

7. what are the consequences of the bias? The author mentioned clinically significant, insignificant, but would be good to know the details. for instance, what would a high HbA1C reading lead to?

8. are the study population the intended patient groups? The inclusion, exclusion criteria are not clear. More details, such as the protocol, would help, laying out the essential PICO question the patients, the user, the comparator, the outcomes for the study.

6. PLOS authors have the option to publish the peer review history of their article (what does this mean?). If published, this will include your full peer review and any attached files.

**Do you want your identity to be public for this peer review?** For information about this choice, including consent withdrawal, please see our Privacy Policy.

Reviewer #1: No

---

## [Decision Letter · Decision Letter 1]

10 Jun 2024

PGPH-D-24-00096R1

Performance and usability of cardiometabolic point of care devices in Nepal: a prospective, quantitative, accuracy study

Dear Dr. Giachino,

Thank you for submitting your manuscript to PLOS Global Public Health. After careful consideration, we feel that it has merit but does not fully meet PLOS Global Public Health’s publication criteria as it currently stands. Therefore, we invite you to submit a revised version of the manuscript that addresses the points raised during the review process.

The manuscript has been evaluated by one reviewer, and their comments are available below. The reviewer has raised additional questions about the methodology and study design. Could you please revise the manuscript to carefully address the concerns raised?

We look forward to receiving your revised manuscript.

Kind regards,

Johanna Pruller, Ph.D.

PLOS Staff Editor

Journal Requirements:

Additional Editor Comments (if provided):

Reviewers' comments:

Reviewer's Responses to Questions

**Comments to the Author**

1. If the authors have adequately addressed your comments raised in a previous round of review and you feel that this manuscript is now acceptable for publication, you may indicate that here to bypass the “Comments to the Author” section, enter your conflict of interest statement in the “Confidential to Editor” section, and submit your "Accept" recommendation.

Reviewer #2: (No Response)

2. Does this manuscript meet PLOS Global Public Health’s publication criteria? Is the manuscript technically sound, and do the data support the conclusions? The manuscript must describe methodologically and ethically rigorous research with conclusions that are appropriately drawn based on the data presented.

Reviewer #2: Yes

3. Has the statistical analysis been performed appropriately and rigorously?

Reviewer #2: Yes

4. Have the authors made all data underlying the findings in their manuscript fully available (please refer to the Data Availability Statement at the start of the manuscript PDF file)?

Reviewer #2: Yes

5. Is the manuscript presented in an intelligible fashion and written in standard English?

Reviewer #2: Yes

6. Review Comments to the Author

Reviewer #2: With interest I read the paper by Giochino et al. describing results of a cross-sectional accuracy study of prospectively recruited participants examining the performance of two different point-of-care devices assessing glucose, total cholesterol and glycated hemoglobin. While the concept is not novel, this manuscript provides information on two "new" devices in a distinct patient population from Nepal.

Overall, the manuscript is well written, the methodology seems robust, and the statistical analyses are appropriately done and presented. Results are comprehensive and detailed.

I have some minor additional comments to the authors to improve their manuscript:

1. Please detail the time elapsed between venous blood collection and laboratory assessment of the markers of interest. This is especially important for glucose, as the longer the wait is, the longer chances of glucose decay is. Also, were all analyses performed in one core lab? several sites? If so what was the coefficient of variance of the machines used for analyses in each site?

2. To support agreement for continuous measurements between different devices/methods, please add an analysis for interclass correlation.

3. Were repeat measurements for intraclass correlation performed? Reliability is another important aspect not highlighted in the current manuscript.

4. Please discuss and compared the results to other existing devices as this field is saturated in widely available and relatively cheap devices.

5. Discussion should also focus on the the gap which these devices are meant to bridge. The ability to perform point-of-care measurements of glucose and cholesterol is not novel. Are the devices in focus cheaper than alternatives? Are they easier to use?

7. PLOS authors have the option to publish the peer review history of their article (what does this mean?). If published, this will include your full peer review and any attached files.

**Do you want your identity to be public for this peer review?** For information about this choice, including consent withdrawal, please see our Privacy Policy.

Reviewer #2: No

---

## [Decision Letter · Decision Letter 2]

4 Sep 2024

Performance and usability of cardiometabolic point of care devices in Nepal: a prospective, quantitative, accuracy study

PGPH-D-24-00096R2

Dear Ms Giachino,

We are pleased to inform you that your manuscript 'Performance and usability of cardiometabolic point of care devices in Nepal: a prospective, quantitative, accuracy study' has been provisionally accepted for publication in PLOS Global Public Health.

Best regards,

Julia Robinson

Executive Editor

Reviewer Comments (if any, and for reference):

Reviewer's Responses to Questions

**Comments to the Author**

1. If the authors have adequately addressed your comments raised in a previous round of review and you feel that this manuscript is now acceptable for publication, you may indicate that here to bypass the “Comments to the Author” section, enter your conflict of interest statement in the “Confidential to Editor” section, and submit your "Accept" recommendation.

Reviewer #1: All comments have been addressed

2. Does this manuscript meet PLOS Global Public Health’s publication criteria? Is the manuscript technically sound, and do the data support the conclusions? The manuscript must describe methodologically and ethically rigorous research with conclusions that are appropriately drawn based on the data presented.

Reviewer #1: Yes

3. Has the statistical analysis been performed appropriately and rigorously?

Reviewer #1: Yes

4. Have the authors made all data underlying the findings in their manuscript fully available (please refer to the Data Availability Statement at the start of the manuscript PDF file)?

Reviewer #1: (No Response)

5. Is the manuscript presented in an intelligible fashion and written in standard English?

Reviewer #1: Yes

6. Review Comments to the Author

Reviewer #1: please update figure 1 which cannot be viewed.

7. PLOS authors have the option to publish the peer review history of their article (what does this mean?). If published, this will include your full peer review and any attached files.

**Do you want your identity to be public for this peer review?** For information about this choice, including consent withdrawal, please see our Privacy Policy.

Reviewer #1: No
